# Comparative Transcriptome-Based Mining of Genes Involved in the Export of Polyether Antibiotics for Titer Improvement

**DOI:** 10.3390/antibiotics11050600

**Published:** 2022-04-29

**Authors:** Xian Liu, Yuanting Wu, Xiaojie Zhang, Qianjin Kang, Yusi Yan, Linquan Bai

**Affiliations:** 1State Key Laboratory of Microbial Metabolism, Shanghai-Islamabad-Belgrade Joint Innovation Center on Antibacterial Resistances, School of Life Sciences & Biotechnology, Shanghai Jiao Tong University, Shanghai 200240, China; liuxian19950811@sjtu.edu.cn (X.L.); yuanting__wu@163.com (Y.W.); xiaojiezhang0831@163.com (X.Z.); qjkang@sjtu.edu.cn (Q.K.); 2Joint International Research Laboratory of Metabolic & Developmental Sciences, Shanghai Jiao Tong University, Shanghai 200240, China; 3Institute of Biopharmaceuticals, Taizhou University, Taizhou 318000, China

**Keywords:** *Streptomyces*, comparative transcriptome, polyether antibiotics, salinomycin, exporter genes

## Abstract

The anti-coccidiosis agent salinomycin is a polyether antibiotic produced by *Streptomyces albus* BK3-25 with a remarkable titer of 18 g/L at flask scale, suggesting a highly efficient export system. It is worth identifying the involved exporter genes for further titer improvement. In this study, a titer gradient was achieved by varying soybean oil concentrations in a fermentation medium, and the corresponding transcriptomes were studied. Comparative transcriptomic analysis identified eight putative transporter genes, whose transcription increased when the oil content was increased and ranked top among up-regulated genes at higher oil concentrations. All eight genes were proved to be positively involved in salinomycin export through gene deletion and trans-complementation in the mutants, and they showed constitutive expression in the early growth stage, whose overexpression in BK3-25 led to a 7.20–69.75% titer increase in salinomycin. Furthermore, the heterologous expression of *SLNHY_0929* or *SLNHY_1893* rendered the host *Streptomyces lividans* with improved resistance to salinomycin. Interestingly, *SLNHY_0929* was found to be a polyether-specific transporter because the titers of monensin, lasalocid, and nigericin were also increased by 124.6%, 60.4%, and 77.5%, respectively, through its overexpression in the corresponding producing strains. In conclusion, a transcriptome-based strategy was developed to mine genes involved in salinomycin export, which may pave the way for further salinomycin titer improvement and the identification of transporter genes involved in the biosynthesis of other antibiotics.

## 1. Introduction 

Polyether antibiotics, also called polyether ionophores, are a broad class of natural compounds produced by actinomycetes, with the vast majority being derived from the genera *Streptomyces* and *Actinomadura* [1]. In recent years, with the discovery of over 120 novel molecules, these chemicals have received more and more attention. Typical polyether antibiotics, including salinomycin, nigericin, lasalocid, and monesin (Figure 1), feature 2–5 ether oxygen atoms and a carboxyl group [2]. This structure enables them to chelate with metal cations, such as Na^+^ and K^+^, and protons to form neutral coordination compounds, which cross the cell membrane and subsequently change ion gradients and osmotic pressures, thus resulting in cell death [1]. Salinomycin, a polyether antibiotic produced by *Streptomyces albus* DSM41398 and its derived strains [3], is widely applied in husbandry because it has properties that kill Gram-positive bacteria and coccidia [4]. Recent studies have found that salinomycin also inhibits the growth of leukemia stem cells [5] and epithelial cancer stem cells [6], indicating that it is a potential anti-tumor drug [6].

The high-titer *Streptomyces albus* strain BK3-25 produces 18 g/L salinomycin under lab conditions [7], but the intracellular accumulation of salinomycin poses a threat to cell growth, which can be released by the strain’s resistance ability [8,9,10]. However, the mechanism underlying this antibiotic resistance remains elusive. According to our previous work, *SLNHY_261* (*slnTII*) and *SLNHY_262* (*slnTI*) in a salinomycin biosynthetic gene cluster (BGC), encoding the ATP-binding subunit and transmembrane subunit, respectively [11], were thought to form an ATP-binding cassette (ABC) complex participating in salinomycin export. When *slnTI* and *slnTII* were deleted in *Streptomyces albus* XM 211, the salinomycin titers of the corresponding mutants declined by only 27.2% and 45.4%, respectively [12], indicating that there are additional genes involved in salinomycin export, which may be located beyond the salinomycin BGC regions.

Actinomycetes have large-capacity transporter protein systems, which participate in cell metabolism, intercellular communication, biosynthesis, and proliferation [13]. The ABC superfamily [14] and major facilitator superfamily (MFS) [15] are two well-studied classes of transporters. Whole-genome sequencing has illustrated that there are numerous ABC and MFS transporter genes in an actinomycetal genome both inside and outside of secondary metabolite BGCs [13,16]. Wang et al. analyzed transcriptome expression differences with expression profile chips and discovered 13 candidate transporter genes outside the natamycin BGC from *Streptomyces chattanoogensis* L10 [17]. Chu et al. built a step-by-step workflow based on the TCDB database BLAST, and they included substrate analysis, transporter classification analysis, and phylogenetic analysis to mine BGC-independent exporters. Together with a tunable plug-and-play exporter module with replaceable promoters and ribosome-binding sites, they realized the titer improvement of macrolide biopesticides in different *Streptomyces* producers [18]. Nevertheless, the current commonly used approaches are mainly sequence-dependent or based on known exporters, and the methods of BGC-independent exporter mining still need development.

Soybean oil serves as the main carbon source in salinomycin fermentation, supplying energy through primary metabolism and precursors, such as malonyl-CoA, methylmalonyl-CoA, and ethylmalonyl-CoA, for salinomycin biosynthesis. Usually, 15% (*w*/*v*) of soybean oil is added to the fermentation medium, which is extremely high compared with other antibiotic fermentations. Our previous work revealed that increased soybean oil addition resulted in higher salinomycin production [19], and, thus, we hypothesized that higher concentrations of soybean oil cause a higher transcription of exporter genes. 

Herein, a strategy based on comparative transcriptomic analysis under different concentrations of soybean oil supplementation was developed to identify salinomycin exporter genes (Appendix A). Our work provided universal exporters for the titer improvement of polyether antibiotics in *Streptomyces*. Furthermore, our method might broaden transporter engineering toolkits for the titer improvement of other valuable products in *Streptomyces*.

## 2. Materials and Methods

### 2.1. Strains, Plasmids, and Culture Conditions

The bacterial strains and plasmids used in this study are listed in Appendix A. 

*S. albus* BK3-25 (from Zhejiang Shenghua Biok Biology Co., Ltd., Deqing, China) and its mutants were grown on ISP4 medium (10 g/L soluble starch, 2 g/L (NH_4_)_2_SO_4_, 1 g/L K_2_HPO_4_, 2 g/L CaCO_3_, 1 g/L NaCl, 1 g/L MgSO_4_, 100 μL of trace element solution (1% ZnSO_4_, 1% MnCl_2_, 1% FeSO_4_ (*w*/*v*)), 20 g/L agar) for 7 days for sporulation. The conjugation of *Streptomyces* with *Escherichia coli* was carried out on ISP4 plates supplemented with 20 mM MgCl_2_. The fermentation procedure was as follows: *S. albus* BK3-25 and its mutants were grown in 30 mL of TSBY medium [20] (30 g/L tryptone soya broth, 5 g/L yeast extract, 103 g/L sucrose) at 30 °C and 220 r.p.m. for 48 h. Then, 1 mL of the culture was transferred into 30 mL of the seed medium (30 g/L soybean meal, 10 g/L yeast extract, 2 g/L CaCO_3_, 80 mL/L 50% glucose) and cultivated at 33 °C and 220 r.p.m. for 16 h. Finally, 5 mL of the seed culture was transferred into 50 mL of the fermentation medium (8 g/L germ powder, 5 g/L soybean meal, 2.2 g/L KCl, 1 g/L NaCl, 1.6 g/L urea, 2 g/L tartaric acid, 0.1 g/L MgSO_4_, 0.1 g/L K_2_HPO_4_, 5 g/L CaCO_3_, pH 6.6–6.9, supplemented with 7.5 g/50 mL soybean oil) and cultured at 33 °C and 220 r.p.m. for 9 days [11]. 

*S. lividans* TK24, *S. cinnamonensis* ATCC 15413, *S. lasaliensis* ATCC 31180, *S. hygroscopicus* XM201-*ga32* and their mutants were grown on SFM medium (20 g/L soybean meal, 20 g/L mannitol, 20 g/L agar) for 7 days for sporulation. The conjugation of *Streptomyces* with *E. coli* was carried out on SFM plates supplemented with 20 mM MgCl_2_. 

For fermentation, *S. cinnamonensis* ATCC 15413 and its mutants were grown in 25 mL of the seed medium (20 g/L dextrin, 15 g/L soybean meal, 2.5 g/L yeast extract, 5 g/L glucose, 1 g/L CaCO_3_, pH 6.7–6.8), and then 2.5 mL of the seed culture was transferred into 25 mL of the fermentation medium (20 g/L soybean oil, 45 g/L glucose, 40 g/L soybean meal, 2.2 g/L NaNO_3_, 2.2 g/L Na_2_SO_4_, 0.07 g/L Al_2_(SO_4_)_3_, 0.1 g/L FeSO_4_, 0.33 g/L MnCl_2_, 0.075 g/L K_2_HPO_4_, 2.5 g/L CaCO_3_, pH 6.7–6.8) and cultured at 32 °C and 250 r.p.m. for 10 days. 

*S. lasaliensis* ATCC 31180 and its mutants were grown in 25 mL of the seed medium (20 g/L sucrose, 20 g/L soybean meal, 5 g/L tryptone, 5 g/L malt extract, 2 g/L NaCl, 4 g/L CaCO_3_, pH 7.0), and then 2.5 mL of the seed culture was transferred into 25 mL of the fermentation medium (5 g/L glucose, 40 g/L dextrin, 35 g/L soybean meal, 7.5 g/L corn starch, 3 g/L NaCl, 4 g/L KH_2_PO_4_, 2 g/L MgSO_4_·7H_2_O, pH 7.0) and cultured at 28 °C and 200 r.p.m. for 6 days [21].

*S. hygroscopicus* XM201-*ga32* and its mutants were grown in 50 mL of the seed medium (10 g/L glucose, 10 g/L tryptone, 5 g/L yeast extract), and then 7.5 mL of the seed culture was transferred into 50 mL of the fermentation medium (30 g/L corn starch, 70 g/L glucose, 40 g/L soybean meal, 3 g/L (NH_4_)_2_SO_4_, 0.01 g/L CoCl_2_, 10 g/L CaCO_3_, 1 g/L soybean oil, pH 6.8–7.0) and cultured at 30 °C and 220 r.p.m. for 7 days [22].

*E. coli* ET12567 (pUZ8002) was used for conjugation. The *E. coli* cells were cultured in Luria–Bertani (LB) broth at 37 °C.

### 2.2. Transcriptome Sequencing of BK3-25

For transcriptome sequencing, mycelia were harvested on the third day of fermentation. The total RNA was extracted using Redzol according to the manufacturer’s instructions. Transcriptome sequencing was performed by the Shanghai Biotechnology Corporation, and the expression level of each gene was calculated as fragments per kilobase of exon per megabase of library size (FPKM). 

### 2.3. Construction of Plasmids for Deletion and Over-Expression of Eight Candidate Genes

For gene deletion through homologous recombination, left and right flanking regions of each gene were obtained using PCR amplification, ligated to *Eco*RV-digested pBluescript SK, and sequenced (Appendix A). Then, these plasmids were digested with *Xba*I/*Eco*RI or *Eco*RI/*Hin*dIII and ligated to *Xba*I/*Hin*dIII-digested plasmid pJTU1278. The primers used for gene deletion are listed in Appendix A.

For gene overexpression, eight candidate genes were obtained using PCR amplification, ligated to *Eco*RV-digested pBluescript SK, and sequenced. Then, these plasmids were digested with *Xba*I/*Not*I (*Nde*I/*Eco*RI), and the fragments containing genes were ligated with the *Xba*I/*Not*I-digested plasmid pIB139 or *Nde*I/*Eco*RI-digested plasmid pLQ646 (Appendix A). The primers used for gene over-expression are listed in Appendix A.

### 2.4. Conjugation between Streptomyces and E. coli

All the plasmids were successively introduced into the non-methylating *E. coli* strain ET12567 (pUZ8002) and *S. albus* strains. Spores (~10^9^ CFU) were heat-shocked at 50 °C for 10 min, pregerminated for 2.5 h, and then mixed with *E. coli* cells. The suspensions were spread onto non-selective plates containing ISP4 medium supplemented with 20 mM MgCl_2_. Apramycin was overlaid on the plates after 17 h of incubation at 30 °C, and exconjugants typically appeared after 3 days. 

For gene deletion, the exconjugants were assessed as single-crossover mutants using PCR amplification with primers SLNHY_X-YZ-F/R (Appendix A). After two rounds of sporulation without antibiotic selection, double-crossover mutants were verified using PCR with primers SLNHY_X-YZ-F/R (Appendix A). For gene over-expression, the exconjugants were verified using PCR with primers pIB139-over-YZ-F (pLQ648-over-YZ-F) and SLNHY_X-over-YZ-R (Appendix A). 

### 2.5. HPLC Analysis of Antibiotics

For the detection of total salinomycin, 1 mL of fermentation broth was mixed with 9 mL of methanol, followed by sonication at 40 kHz for 30 min. Then, 1 mL of the mixture was taken and centrifuged at 12,000 r.p.m. for 1 min, and the supernatant was filtrated and subjected to HPLC analysis. For the detection of intracellular salinomycin, 1 mL of the fermentation broth was centrifuged at 12,000 r.p.m. for 5 min, and the supernatant was discarded. The mycelia were washed twice with water and mixed with 1 mL of methanol, followed by sonication at 40 kHz for 30 min. Then, the mixture was centrifuged at 12,000 r.p.m. for 5 min, and the supernatant was filtrated and subjected to HPLC analysis. HPLC was performed on Agilent series 1260 (Agilent Technologies, Santa Clara, CA, USA) with an Agilent TC-18 column (2.1 × 150 mm, 5 μm). In this process, 8% A (water, 2% acetic acid) and 92% B (acetonitrile) were used as the mobile phase with a flow rate of 1 mL/min, and the detection time was 20 min using UV spectroscopy at 210 nm [11]. The concentrations of salinomycin were calculated according to the standard curve of salinomycin (Appendix A).

For the detection of lasalocid, 1 mL of fermentation broth was mixed with 1 mL of methanol, followed by sonication at 40 kHz for 30 min. Then, the mixture was centrifuged at 12,000 r.p.m. for 1 min, and the supernatant was filtrated and subjected to HPLC analysis. HPLC was performed on Agilent series 1260 (Agilent Technologies, USA) with an Agilent TC-18 column (2.1 × 150 mm, 5 μm) at 40 °C. In this process, 15% A (water, 0.125 mol/L ammonium acetate, pH 4.8) and 85% B (acetonitrile) were used as the mobile phase with a flow rate of 1 mL/min, and the detection time was 20 min using UV spectroscopy at 305 nm [21]. The concentrations of lasalocid were calculated according to the standard curve of lasalocid (Appendix A).

For the detection of monensin, 1 mL of fermentation broth was centrifuged at 12,000 r.p.m. for 5 min, and the supernatant was discarded. The mycelia were washed twice with water and mixed with 1 mL of ethanol, followed by sonication at 40 kHz for 30 min. Then, the mixture was centrifuged at 12,000 r.p.m. for 5 min, and the supernatant was filtrated and subjected to HPLC analysis. HPLC was performed on Agilent series 1260 (Agilent Technologies, Santa Clara, CA, USA) with an Agilent TC-18 column (2.1 × 150 mm, 5 μm). In this process, 20 mM ammonium acetate and methanol were used as the mobile phase with a flow rate of 1 mL/min. For 0–25 min, the ratio of methanol increased from 80% to 100%, and for 25–30 min, it decreased from 100% to 80%. Evaporative light-scattering detection (ELSD) was conducted for 30 min at 85 °C [23]. The concentrations of monensin were calculated according to the standard curve of monensin (Appendix A).

For the detection of nigericin, 1 mL of fermentation broth was mixed with 9 mL of methanol, followed by sonication at 40 kHz for 30 min. Then, 1 mL of the mixture was taken and centrifuged at 12,000 r.p.m. for 1 min, and the supernatant was filtrated and subjected to HPLC analysis. HPLC was performed on Agilent series 1260 (Agilent Technologies, USA) with an Agilent TC-18 column (2.1 × 150 mm, 5 μm). A gradient elution (1 mL/min flow rate) was performed using A (methanol/water, 9:1 ratio, 0.1% TFA from 0 to 25 min) and B (methanol, 100%, 0.1% TFA, from 25 to 50 min). ELSD detection was conducted for 50 min at 85 °C [24]. The concentrations of nigericin were calculated according to the standard curve of nigericin (Appendix A).

### 2.6. RNA Extraction and RT-qPCR Analysis

Mycelia of *S. albus* BK3-25 were harvested, and the total RNA was extracted using Redzol according to the manufacturer’s instructions (SBS Genetech, Shanghai, China) [25]. The quality of the RNA was determined using a NanoDrop 2000 spectrophotometer. For RT-qPCR experiments, total RNA was reversely transcribed into cDNA using RevertAid^TM^ H Minus First Strand cDNA Synthesis Kit (Thermo Fisher, Waltham, MA USA). The RT-qPCR experiments were carried out on a 7500 Fast Real-time RCR system (Applied Biosystems, Waltham, MA USA) using Maxima^TM^ SYBR Green/ROX qPCR Maxter Mix (Thermo Fisher, Waltham, MA USA) according to the manufacturer’s procedure. The expression values of the target genes were calculated using 2^−ΔΔCT^ methods with the housekeeping gene *hrdB* as internal control [26].

### 2.7. Biomass Determination under Fermentation Condition

Due to the insoluble residues in the liquid medium, total intracellular nucleic acid rather than dry cell weight was determined to represent the growth of *Streptomyces*. The concentration of intracellular nucleic acid was detected as follows: 1 mL fermentation broth was centrifuged and washed twice to eliminate the interference of the medium. Then, 1 mL of Solution A (1.5 g diphenylamine, 100 mL acetic acid, 1.5 mL concentrated sulfuric acid, 1 mL 1.6% acetaldehyde) was mixed with the mycelia and put in water bath at 60 °C for 1 h. Then, the mixture was centrifuged, and 150 μL of supernatant was transferred into 96-well plates and detected at 595 nm (Infinite M200 PRO, TECAN, Männedorf, Switzerland).

## 3. Results

### 3.1. Transcriptome-Based Identification of Candidate Exporter Genes

We initially assumed that the higher the salinomycin titer, the higher the transcription of the involved exporter genes. In order to establish a titer gradient, different concentrations of soybean oil (5%, 10%, and 15%) were supplemented to the fermentation broth, and the corresponding salinomycin titers were 5.30 g/L, 12.50 g/L, and 17.40 g/L, respectively. Using these three samples, transcriptomic data were collected using RNA-seq technology. According to the hypothetic concurrent relationship between salinomycin titers and the transcription of exporter genes, eight exporter genes with increasing expression patterns and that topped the fold-change of transcription at a higher salinomycin titer were selected from 248 ABC transporter genes and 23 MFS genes in the BK3-25 genome. These eight genes include seven ABC transporter genes, *SLNHY_3363*, *SLNHY_4037*, *SLNHY_6316*, *SLNHY_6652*, *SLNHY_0818*, *SLNHY_0199*, and *SLNHY_1893*, and one MFS gene, *SLNHY_0929* (Figure 2 and Appendix A). Although the transcriptions of *SLNHY_3363* and *SLNHY_0199* at 15% oil supplementation were lower than those at 10%, both of their transcriptions dramatically rose when the oil contents were shifted from 5% to 10%. 

In order to verify the transcriptomic data, the transcription levels of the eight candidate genes were measured using RT-qPCR with cultures collected on the third day of fermentation supplemented with 5% or 15% soybean oil, and all genes showed higher expressions with 15% oil supplementation, which was consistent with the transcriptomic data. Among them, *SLNHY_929* demonstrated the highest expression, followed by *SLNHY_3363* and *SLNHY_1893* (Appendix A). 

### 3.2. All Eight Candidate Genes Were Positively Involved in Salinomycin Export

To investigate whether these eight genes are involved in salinomycin production, they were knocked out through homologous recombination. As shown in Figure 3A,B, the total salinomycin titers of all mutants decreased to 11.62–27.36% of that of BK3-25, and the intracellular concentrations of salinomycin increased to 143.61–237.89% of that of BK3-25, indicating that these genes were positively related to salinomycin biosynthesis. Moreover, the deletion of Δ*SLNHY_0199* was the most pronounced, with a dramatic decrease in the salinomycin titer from 13.34 g/L to 1.55 g/L.

Further verification of the above conclusion was conducted through trans-complementation of each mutant with the corresponding gene cloned under the control of *PermE**. All individually complemented strains returned to 78.77–88.77% of the original titer of salinomycin (Figure 3C), and the intracellular accumulations of salinomycin synchronically returned to 62.36–105.19% of the level of BK3-25 (Figure 3D), providing more proof of the involvement of these eight genes in salinomycin biosynthesis and, most likely, in its export. 

In addition, these eight genes were individually over-expressed in BK3-25 to see whether they played vital roles in salinomycin titer improvement. As expected, compared with the control strain bearing the empty vector pIB139, the excessive expression of these genes all increased salinomycin titers by 7.20–69.75%, especially BK3-25::*SLNHY_3363* and BK3-25::*SLNHY_0929*, which had improved titers of 24.60 g/L and 22.85 g/L, respectively (Figure 3E). Accordingly, the intracellular salinomycin accumulations of all mutants showed a marked fall to 24.35–46.23% of the same level of BK3-25 (Figure 3F). 

### 3.3. These Eight Exporter Genes Were Constitutively Expressed

In order to determine whether the expressions of these eight genes were constitutive or induced by salinomycin, the transcription profiles of each gene were obtained using RT-qPCR with samples collected each day during the whole fermentation period (Figure 4). Compared with the house-keeping gene *hrdB*, these exporter genes were actively transcribed at the very beginning and then gradually decreased along with the fermentation process. Since salinomycin obviously accumulated after the first day, as shown in Appendix A, we can safely draw the conclusion that these genes were constitutively expressed rather than being induced by salinomycin.

### 3.4. SLNHY_0929 and SLNHY_1893 Improved Resistance to Salinomycin in Streptomyces lividans

Even though these eight exporter genes were proved to be involved in salinomycin export, we still needed to determine why they functioned in this way. *Streptomyces lividans* TK24 was found to be susceptible to high concentrations of salinomycin, and the minimal inhibition concentration (MIC) was 0.5 mmol/L for the control strain TK24::pIB139. These eight genes were individually introduced into *S. lividans* TK24. Although most mutants carrying the introduced exporter genes maintained similar susceptibility to salinomycin, TK24::*SLNHY_0929* and TK24::*SLNHY_1893* rendered the host with an improved resistance as high as 1.0 mmol/L (Figure 5A,B). These data strongly suggest the salinomycin export ability of *SLNHY_0929* and SLNHY_1893, which exported the assimilated exogenous salinomycin out of *S. lividans* TK24.

### 3.5. SLNHY_0929 Was a Universal Exporter for Polyether Antibiotics with Similar Structure with Salinomycin

Since most polyether antibiotics shared similar hydrophobic structures, we wondered whether the exporter genes were universal in pumping them out and whether the genes played roles in improving their titers. Therefore, the three exporter genes with the most substantial effects on salinomycin titers when over-expressed in BK3-25, i.e., *SLNHY_0929*, *SLNHY_3363* and, *SLNHY_**4037*, were heterologously expressed in *Streptomyces lasaliensis* ATCC 31180 (a lasalocid producer), *Streptomyces cinnamonensis* ATCC 15413 (a monensin producer), and *Streptomyces hygroscopicus* XM201-*ga*32 (a nigericin producer). Herein, the previously used *PermE** promoter was replaced by a stronger promoter *kasO*p*, since the latter was reported to work better in XM201 [22,27].

Interestingly, the heterologous expression of *SLNHY_0929* resulted in a significant improvement in all three antibiotics, with lasalocid titers from 163.60 mg/L to 262.50 mg/L in *S. lasaliensis* (Figure 6A), monensin titers from 572.40 mg/L to 1,286.00 mg/L in *S. cinnamonensis* (Figure 6B), and nigericin titers from 116.77 mg/L to 207.27 mg/L in *S. hygroscopicus* (Figure 6C). Surprisingly, the heterologous expressions of *SLNHY_3363* and *SLNHY_**4037* had no positive effects on the production of these three polyether antibiotics, even with unexpected, dropped titers, and the reason needed further exploration. Overall, these results clearly show that *SLNHY_0929* is a universal exporter for salinomycin, lasalocid, monensin, and nigericin, which shared similar molecular structures.

Transporter engineering has been considered as a promising strategy to maximize secondary metabolite production in bacterial hosts, such as *Streptomyces* spp. and *Aspergillus* spp. [28,29]. Except for the exporters located in BGCs, BGC-independent exporters caused by horizontal gene transfer may also contribute to metabolite export [30,31]. To discover transporter genes located far from BGCs, expression profiling, genome-wide knockout studies, stress-based selection, and the inhibitor strategy have often been used [32]. 

Herein, due to *Streptomyces albus*’ highly efficient utilization of soybean oil, we found that salinomycin production rose as oil addition rose, and the transcription levels of the genes involved in salinomycin PKS, β-oxidation, and precursor biosynthesis also increased [7]. Since transporters are the essential channels of both precursor import and salinomycin export, they are more necessary with the rise in ionophore product synthesis, so we decided to focus on the study of transporters. Next, we managed to mine eight transporter genes outside salinomycin BGC based on comparative transcriptome data under different salinomycin titers. Furthermore, all eight selected genes proved to be correlated with salinomycin synthesis by gene deletion and over-expression. Among the eight genes, only *SLNHY_0929* and *SLNHY_1893* encoded proteins that could pump salinomycin out of the cell and, thus, provide self-resistance to the host, according to the heterologous expression in the model strain *S. lividans* and the salinomycin supplement experiments. Our work constructed a novel method for antibiotic transporter gene mining. The fermentation characteristics of *Streptomyces albus* with a soybean oil preference were focused and combined with transcriptome sequencing technology, which showed a very successful titer improvement of salinomycin through exporter engineering and might apply to other antibiotics with oil-derived precursors. Therefore, this study also provides an application with great potential in promoting the more cost-effective production of salinomycin and other chemicals.

MFS-type transporters are channels of multiple substrates, such as monose, polysaccharides, amino acids, polypeptides, vitamins, cofactors, secondary metabolites, chromophores, and bases. They can either function independently or cooperate with ABC transporters in metabolite efflux [17]. In our work, the MFS family gene, *SLNHY_0929*, was heterologously expressed in the producers of other polyether antibiotics, and it showed a broad spectrum of substrate identification, whose protein pumped out intracellular salinomycin, lasalocid, monensin, and nigericin, as well as increasing the titers of each. As a matter of fact, *S. cinnamonensis* ATCC 15413 possesses a homolog of the exporter gene *SLNHY_0929*, namely, *orf6552*, with 93% coverage and 75.81% identity, while none of the homolog is present in *S. lasaliensis* ATCC 31180 or *S. hygroscopicus* XM201-*ga32*. Thus, ORF6552 is considered to be an endogenous MFS transporter, which exports monensin from the host. Whether or not it served as another universal ionophore pump remains to be explored. Compared to specific exporters slnTI and slnTII in the gene cluster, *SLNHY_0929* contributed more to salinomycin biosynthesis [12]. Meanwhile, endogenous exporters in lasalocid, monensin, and nigericin BGCs, which provide self-resistance to producing strains, have been reported. Lsd5 in *S. lasaliensis* showed 53% identity with MonT in *S. cinnamonensis*, and in *S. hygroscopicus*, R14/R15 formed an ABC transporter with 71% and 49% identities with the TMD and NBD of the transporter in *S. avermitilis*’s BGC, respectively [33,34,35,36]. Until now, there have been no data about these self-exporters’ functions on antibiotic production, so we could not compare them with the non-specific exporter *SLNHY_0929*. However, introducing this MFS transporter improved the polyether antibiotics’ titers by over 60% and almost doubled monensin’s titer, which demonstrates the efficacy of our method for transporter mining. Due to the constitutive expression of these exporter genes, we can safely draw the conclusion that *SLNHY_0929* served as a universal and stable pump by flexibly identifying polyether compounds. Whether or not it could recognize other types of chemicals remains unknown. SLNHY_1893, SLNHY_3363, SLNHY_4037, SLNHY_0199, SLNHY_0818, SLNHY_6316, and SLNHY_6652 all belong to ABC transporters; however, they cannot export salinomycin according to the MIC results implemented in *Streptomyces lividans*. Their functions were speculated to be importers of soybean oil or other small nutrient molecules, or exporters of other secondary metabolites, such as actinopyranone and elaiophylin, whose BGCs were detected in the *S. albus* genome. Hence, on the one hand, in a future study, intracellular acyl-CoA concentrations will be detected to study whether these proteins pump fatty acids and glycerol derived from soybean oil [37]. On the other hand, we would like to strengthen or weaken the expression of these pumps and observe the production of possible metabolites to identify the pumps’ function. 

Additionally, since each transporter exported salinomycin and improved its biosynthesis, it would be interesting to examine whether these pumps are competitive or cooperative. Thus, tandem over-expression of two or more genes will be employed to study their relationship, which may push the salinomycin titer to a higher level. By means of electronic microscope observation and molecular dynamics simulation, the conformation change of transporters during the pumping of salinomycin will be analyzed. Besides export, other resistance strategies, including the inactivation of antibiotics and the modification of function targets, are worthy of study with regard to salinomycin. Further bioinformatic and functional analyses are likely to provide answers to these questions. 

## 4. Conclusions

In summary, we constructed a novel method for salinomycin exporter mining by combining soybean oil preference and transcriptome analysis. We identified eight BGC-independent transporters and verified their functions. Finally, one of them was proved to be efficient in multiple polyether antibiotic-producing hosts. Our work contributes new strategies to further the improvement of salinomycin titers, and it paves the way for increasing the titers of other antibiotics through transporter engineering.

## Figures and Tables

**Figure 1 antibiotics-11-00600-f001:**
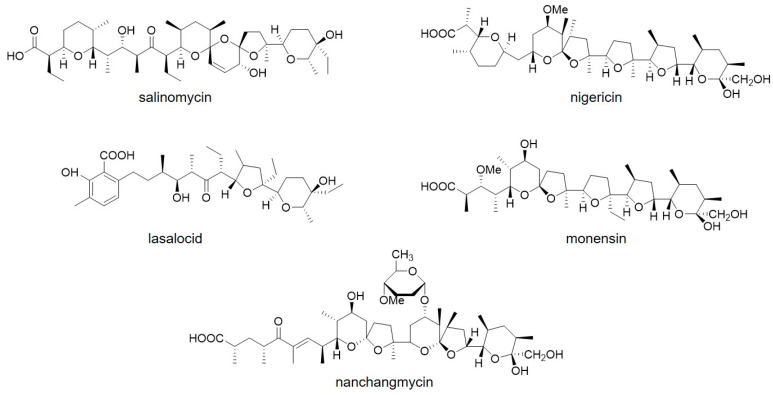
Structures of polyether antibiotics.

**Figure 2 antibiotics-11-00600-f002:**
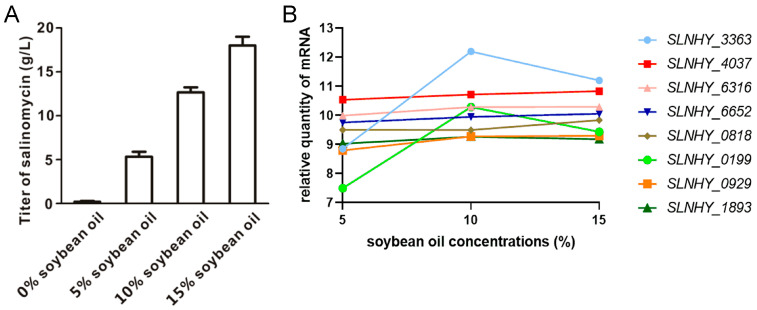
(**A**) Salinomycin titers under 0, 5, 10, and 15% soybean oil supplementation. (**B**) Transcription profiles of eight candidate transporter genes under 5, 10, and 15% (*w*/*v*) soybean oil supplementation.

**Figure 3 antibiotics-11-00600-f003:**
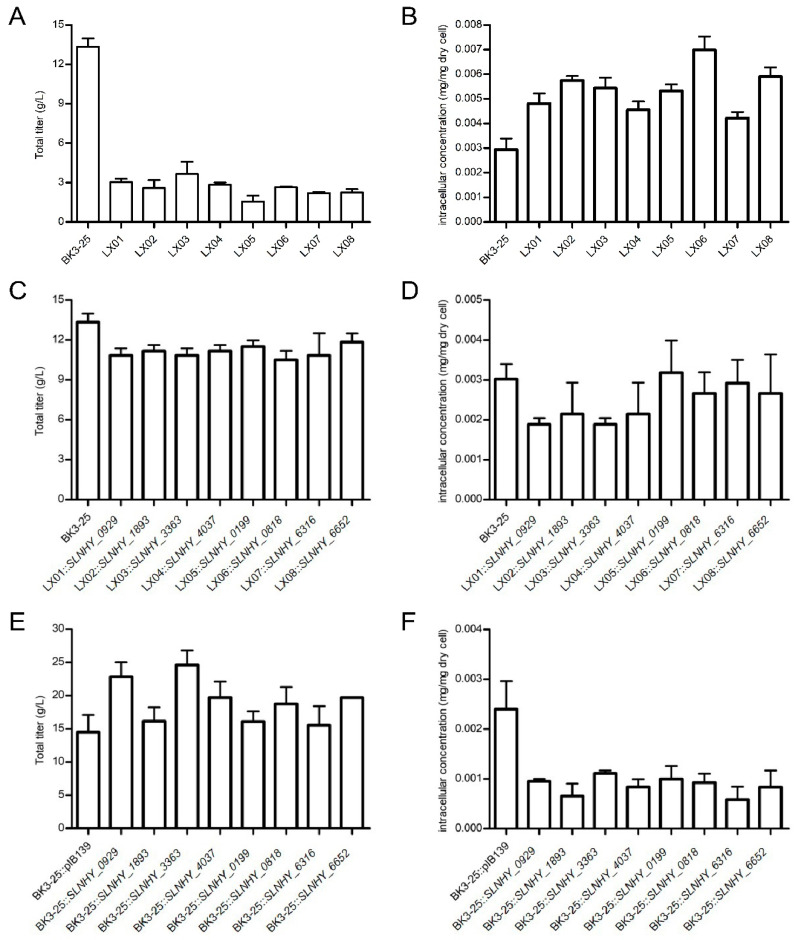
Salinomycin production of exporter gene mutants. (**A**) Total salinomycin titers of gene deletion mutants of BK3-25. (**B**) Intracellular salinomycin concentrations of gene deletion mutants. (**C**) Total salinomycin titers of gene complement mutants. (**D**) Intracellular salinomycin concentrations of gene complement mutants. (**E**) Total salinomycin titers of gene over-expression mutants. (**F**) Intracellular salinomycin concentrations of gene over-expression mutants. The total salinomycin titers were calculated based on the fermentation broth volume (**A**,**C**,**E**), and the intracellular salinomycin concentrations were calculated based on the dry cell weight (**B**,**D**,**F**). LX01, BK3-25Δ*SLNHY_0929*; LX02, BK3-25Δ*SLNHY_1893*; LX03, BK3-25Δ*SLNHY_3363*; LX04, BK3-25Δ*SLNHY_4037*; LX05, BK3-25Δ*SLNHY_0199*; LX06, BK3-25Δ*SLNHY_0818*; LX07, BK3-25Δ*SLNHY_6316*; LX08, BK3-25Δ*SLNHY_6652*.

**Figure 4 antibiotics-11-00600-f004:**
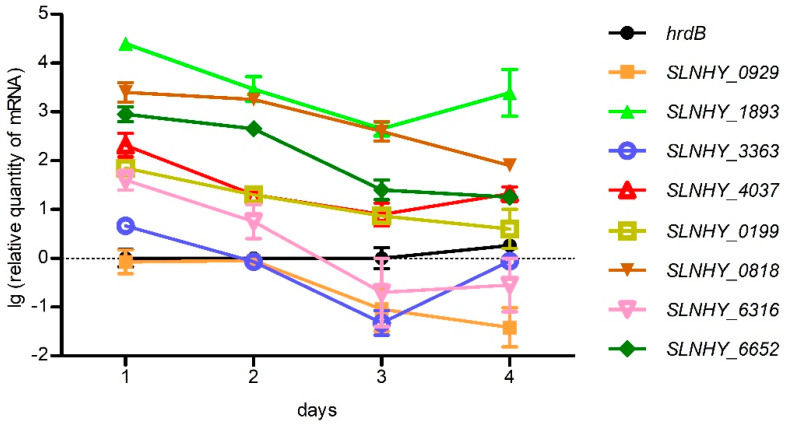
Transcription profiles of eight exporter genes. Line with black circle, *hrdB*; orange square, *SLNHY_0929*; green equilateral solid triangle, *SLNHY_1893*; blue circle, *SLNHY_3363*; red equilateral hollow triangle, *SLNHY_4037*; earthy yellow hollow square, *SLNHY_0199*; brown inverted solid triangle, *SLNHY_0818*; pink inverted hollow triangle, *SLNHY_6316*; dark green rhombus, *SLNHY_6652*. Logarithm of transcription data were taken as ordinate; if not, it would be difficult to show their wide fluctuation scales.

**Figure 5 antibiotics-11-00600-f005:**
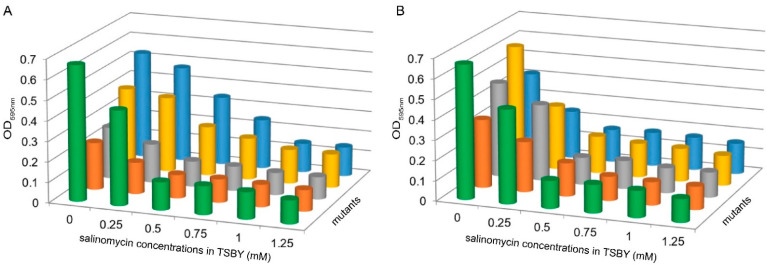
Biomass profiles of *S. lividans* TK24 mutants under different concentrations of salinomycin. Total intracellular nucleic acid was determined to represent the growth of *S. lividans*. (**A**) Green column, TK24::pIB139; orange column, TK24::*SLNHY_3363*; gray column, TK24::*SLNHY_4037*; yellow column, TK24::*SLNHY_0929*; blue column, TK24::*SLNHY_1893*. (**B**) Green column, TK24::pIB139; orange column, TK24::*SLNHY_0199*; gray column, TK24::*SLNHY_0818*; yellow column, TK24::*SLNHY_6316*; blue column, TK24::*SLNHY_6652*.

**Figure 6 antibiotics-11-00600-f006:**
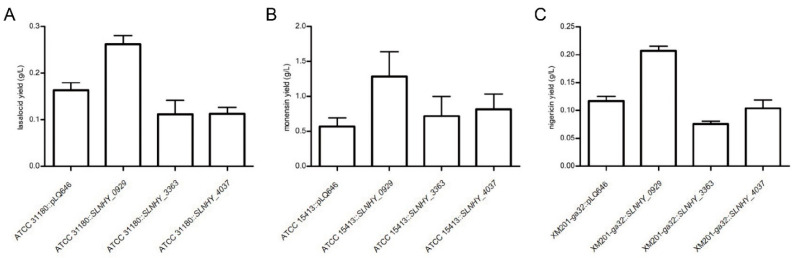
Production of polyether antibiotics with heterologous expressions of exporters in other Streptomyces hosts. (**A**) Lasalocid production of *S. lasaliensis* mutants. (**B**) Monensin production of *S. cinnamonensis* mutants. (**C**) Nigericin production of *S. hygroscopicus* mutants. pLQ646, integrative vector with *kasO*p* promoter.

## Data Availability

Not applicable.

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
