# Peer review of "Comparative Transcriptome-Based Mining of Genes Involved in the Export of Polyether Antibiotics for Titer Improvement"

_antibiotics, 2022, doi:10.3390/antibiotics11050600_

Round 1
Reviewer 1 Report
The manuscript described the comparative transcriptome-based mining of genes involved in the export of polyether antibiotics. The manuscript is well written and executed with good experimental design and results. I recommend the writing to be considered for publication upon addressing these comments:-
1) Consider to revise the title to include the vital impact of the work in increasing the yield of these antibiotics.
2) For the introduction, the authors could consider to include more background information on polyether antibiotics, Streptomyces and Actinobacteria to increase the reader based for this writing.
3) For the method section, it would be really helpful to help readers to understand the work easier, if a Figure could be generated to explain the processes visually.
4) Line: 389-390: Appreciate further clarification on the impact and importance of knowing whether these pumps were competitive, coorperative or complementary to the aim of this project.
5) In the discussion and conclusion, it would be important for authors to highlight the impact of the study, especially the potential application of these findings in promoting better cost effectiveness of producing salinomycin. Also highlighting the huge potential and impact of these findings in the area of drug discovery in the conclusion section.
Reviewer 2 Report
This paper reports on the identification and characterization of the possible exporter of salinomycin, a polyether antibiotic produced by Streptomyces albus. Based on the positive effect of soybean oil on salinomycin yield, the authors studied the change in the transcriptional profile along with the addition of soybean oil and successfully identified eight transporter genes whose transcription levels increased in an oil concentration-dependent manner. Evidence from gene disruption and overexpression experiments supported the involvement of transporters in the secretion of salinomycin. Furthermore, the authors suggest the possible application of the knowledge to improve the production yield of other antibiotics.
This reviewer feels that this study is interesting and well performed. The results are clear and based on reliable data. However, the following points should be considered for an effective revision.
Line 237, Fig. 2B and Table S3
The graph and table show the relative amounts of mRNA, but the reference used for the calculation of the ratio is not designated (probably they are the values when the transcription level of an inner standard gene under 0% soybean oil is defined as 1).
Have the authors checked whether the transcription level of salimomycin biosynthesis genes is affected by the addition of soybean oil?
Lines 256
Exporter gene knockout mutants were successfully generated and shown to have high intracellular salinomycin level. This makes this reviewer wonder why they are not susceptible to the high concentration of salinomycin as S. lividans TK24 is (line 303). Please explain.
Lines 329-337
The high production yield conferred by the introduction of SLNHY_0929 in heterologous hosts is interesting. Are these host strains susceptible to salinomycin? If so, does the introduction of the exporter gene affect susceptibility as did in S. lividadns? Do host strains retain any homolog of the exporter gene? Ionophore production in these organisms also affected by the addition of soybean oil to the medium?
Discussion
This reviewer feels that the Discussion section needs effective revision, including English editing. The 2nd paragraph (lines 351-362) just repeats the description of the results. The authors should describe their idea of why the addition of soybean oil stimulates the transcription of exporter genes resulting in the effective secretion of salinomycin. They may mention the effect of the oil on the transcription of other genes. Explanation regarding the property of an MFS type transporter should be also added.
Supplementary materials
The Word file contains the modification log.
Reviewer 3 Report
Dear Authors,
An interesting publication, written in a clear way despite the complicated layout of the studies, involving transcription of many genes. The subject matter of the paper has an obvious practical dimension. The obtained results can be used in pharmaceutical biotechnology.
The manuscript has a proper layout and the length of particular subchapters does not raise any objections.
The introduction is interesting, although with little emphasis on the purpose of the study. The significance of this work could be emphasized more strongly.
Methods: Described sufficiently, only the conditions for sonification of mycelium are worth supplementing. It is long , 30 minutes, so details of this method should be added.
Results described clearly, despite their high complexity and many dependencies. Graphics are sufficient. Please justify the choice of reference gene in RT-qPCR method.
Discussion of results interesting, develops the topic well.
